

# A time-dependent regularization of the Redfield equation

Antonio D'Abbruzzo[1*], Vasco Cavina[2] and Vittorio Giovannetti[3]

**1** Scuola Normale Superiore, I-56126 Pisa, Italy
**2** Complex Systems and Statistical Mechanics, Physics and Materials Science,
University of Luxembourg, L-1511 Luxembourg, Luxembourg
**3** NEST, Scuola Normale Superiore and Istituto Nanoscienze-CNR, I-56127 Pisa, Italy

\* antonio.dabbruzzo@sns.it

## Abstract

We introduce a new regularization of the Redfield equation based on a replacement of the Kossakowski matrix with its closest positive semidefinite neighbor. Unlike most of the existing approaches, this procedure is capable of retaining the time dependence of the Kossakowski matrix, leading to a completely positive divisible quantum process. Using the dynamics of an exactly-solvable three-level open system as a reference, we show that our approach performs better during the transient evolution, if compared to other approaches like the partial secular master equation or the universal Lindblad equation. To make the comparison between different regularization schemes independent from the initial state, we introduce a new quantitative approach based on the Choi-Jamiołkowski isomorphism.

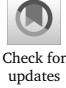

# 1  Introduction

Describing the time evolution of a quantum system interacting with an external environment is of paramount relevance in contemporary physics, with applications in a wide variety of fields. However, portraying the exact dynamics of an open system is often challenging, due to the intrinsic complexity of dealing with the large number of degrees of freedom of the environment. This problem can be tackled by introducing effective descriptions in which only a small, but essential, amount of bath properties is taken into account to derive a good approximated picture of the system evolution. One of such examples are Markovian Quantum Master Equations (QMEs), which express the time derivative of the density matrix in terms of a superoperator satisfying the prescriptions of the Lindblad-Gorini-Kossakowski-Sudarshan (LGKS) theorem [1, 2].

QMEs are one of the most widely used models in open quantum systems and have been applied to problems of quantum transport, computation, chemical modeling and quantum thermodynamics [3, 4]. Despite that, the range of validity of QMEs and their reliability in the description of coherent effects, are still debated today [5]. The standard Born-Markov approximation [6, 7] leads to the Redfield equation [8, 9], which is known to violate the positivity of the density operator [10] and other desirable properties of the quantum evolution [11, 12], hence providing nonphysical predictions. Historically, the main route for curing these issues required the energy levels of the system to be well separated: this is the secular approximation, which provided remarkable results in the context of quantum optics and quantum chemistry [13, 14]. This approach is however not suited for the study of generic many-body systems, where the spacing between energy levels typically decreases exponentially when increasing the size. For this reason, in recent years a number of works appeared in the literature proposing ways to obtain LGKS equations that are free of the restrictions imposed by the secular approximation [15–26].

In this work we argue that these "regularization" techniques amount to a substitution of a certain matrix that describes the dissipative dynamics, known as the Kossakowski matrix, with a positive semidefinite one, thus leading to a LGKS equation. With this observation at hand we propose a natural regularization scheme for the Redfield equation, which consists in replacing the Kossakowski matrix with its closest positive semidefinite matrix of the same dimension. The result is compared with some of the existing schemes, by examining to what extent they reproduce the true dynamics of a simple open system that can be solved exactly. To do this, we employ a novel technique that makes use of the Choi-Jamiołkowski isomorphism [27, 28] to envision a numerical comparison that is independent of the choice of the initial state of the evolution. We emphasize that our procedure can be applied not only to the standard Redfield equation but also to its version with time-dependent coefficients (that can result, for instance, from avoiding the so-called "second Markov" approximation). In this case, our regularization preserves the time dependence of the coefficients but makes the dynamics completely positive divisible [29–31]. The residual time dependence is an indicator that our approach could perform better at short times with respect to existing schemes.

The paper is structured as follows. In Sec. 2 we write the Redfield equation and the associated Kossakowski matrix. In Sec. 3 we discuss the regularization procedure, first by analyzing

common existing schemes in Sec. 3.1 and then by presenting our natural proposal in Sec. 3.2. In Sec. 4 the reader can find an example of application to an open three-level system: in Sec. 4.1 we present a direct numerical calculation at the level of the density matrix, while in Sec. 4.2 we discuss the novel Choi operator technique to compare predictions of different master equations. Finally, in Sec. 5 we draw our conclusions.

## 2 The LGKS theorem and the Redfield equation

Let us consider a quantum system $\mathcal{S}$ described by a Hilbert space $\mathcal{H}_\mathcal{S}$ of finite dimension $N$. Moreover, let $L(\mathcal{H}_\mathcal{S})$ be the vector space of linear operators on $\mathcal{H}_\mathcal{S}$ equipped with the Hilbert-Schmidt inner product $\langle X, Y \rangle := \mathrm{Tr}(X^\dagger Y)$. A linear map $\Phi : L(\mathcal{H}_\mathcal{S}) \to L(\mathcal{H}_\mathcal{S})$ is expected to describe a physical transformation when it is trace-preserving and completely positive (CPT) [6]. A particularly important case is when the process is described by a quantum dynamical semigroup [32], i.e., a one-parameter family $\{\Phi_t\}_{t\geq 0}$ of linear CPT maps on $L(\mathcal{H}_\mathcal{S})$ such that $t \mapsto \Phi_t$ is continuous, $\Phi_0 = \mathbb{1}$, and $\Phi_{t+s} = \Phi_t \circ \Phi_s$. In this case, if $\rho(0)$ is the initial state of $\mathcal{S}$ then the state at time $t$ is given by $\rho(t) = \Phi_t(\rho(0))$. Quantum dynamical semigroups are important because given $\{\Phi_t\}$ we can find a linear operator $\mathcal{L}$, called the generator of the semigroup, such that

$$\frac{\mathrm{d}\rho(t)}{\mathrm{d}t} = \mathcal{L}(\rho(t)), \tag{1}$$

which is in the form of a Markovian master equation [6]. The well-known theorem of Lindblad [1], Gorini, Kossakowski, and Sudarshan [2] characterizes the shape of such a generator. Here we will use the formulation provided by [33], which is best suited for our discussion.

**Theorem.** A linear operator $\mathcal{L} : L(\mathcal{H}_\mathcal{S}) \to L(\mathcal{H}_\mathcal{S})$ is the generator of a quantum dynamical semigroup if it can be written in the form

$$\mathcal{L}(\rho) = -i[H, \rho] + \sum_{i,j=1}^{N^2} \chi_{ij}\left[F_i \rho F_j^\dagger - \frac{1}{2}\left\{F_j^\dagger F_i, \rho\right\}\right], \tag{2}$$

where $H = H^\dagger$, $\{F_i\}_{i=1,\dots,N^2}$ is an orthonormal basis of $L(\mathcal{H}_\mathcal{S})$ and $\chi$ is a positive semidefinite complex matrix which is uniquely determined by the choice of $\{F_i\}$, called Kossakowski matrix.

One could be also interested in generalizations of Eq. (1) in which the generator becomes time dependent $\mathcal{L}_t$. In this kind of scenario we must deal with a two-parameter semigroup $\Phi_{t,s} = \mathcal{T}\exp\left(\int_s^t d\tau \, \mathcal{L}_\tau\right)$, where $\mathcal{T}$ is the time ordering. Here divisibility is guaranteed, in the sense that for any $t \geq s \geq 0$ there exists a linear map $\Lambda_{t,s}$ (intertwining map) such that $\Phi_{t,0} = \Lambda_{t,s} \circ \Phi_{s,0}$. The case of completely positive intertwining maps (CP divisibility) is particularly important, since it is characterized by a LGKS-like generator (2) where $\chi$ and $H$ become time-dependent quantities [29–31]. At this level, notice that $\chi(t) \geq 0$ for all $t \geq 0$ is a sufficient condition for CPT dynamics but it is by no means necessary [15].

In a general setting where $\mathcal{S}$ is allowed to interact with an environment $\mathcal{E}$, the non-Hamiltonian terms in Eq. (2) play a crucial role. This can be seen with a microscopic derivation of (2), in which we start from a unitary description of the universe $\mathcal{U} = \mathcal{S} \cup \mathcal{E}$, trace away the environment and obtain, under suitable assumptions, a master equation for $\mathcal{S}$ which is in LGKS form. Since the universe is closed by definition, it is described by a Hamiltonian, which is commonly written as

$$H_\mathcal{U} = H_\mathcal{S} \otimes \mathbb{1}_\mathcal{E} + \mathbb{1}_\mathcal{S} \otimes H_\mathcal{E} + H_I. \tag{3}$$

The interaction term $H_I$ is of the form

$$H_I = \sum_{\alpha=1}^{M} A_\alpha \otimes B_\alpha \,, \tag{4}$$

where $A_\alpha$ acts on the system and $B_\alpha$ acts on the environment. It is also common to assume these coupling operators to be Hermitian, but here we will not make this assumption.

Under the Born-Markov approximation (and other standard assumptions) it is known that the reduced dynamics of $\mathcal{S}$ in the interaction picture $\widetilde{\rho}(t) := e^{iH_S t}\rho(t)e^{-iH_S t}$ is [6]

$$\frac{d\widetilde{\rho}(t)}{dt} = \sum_{\alpha,\beta} \int_0^t d\tau\, c_{\alpha\beta}(\tau)[\widetilde{A}_\beta(t-\tau)\widetilde{\rho}(t), \widetilde{A}_\alpha^\dagger(t)] + \text{H.c.}\,, \tag{5}$$

where $c_{\alpha\beta}(\tau) := \langle \widetilde{B}_\alpha^\dagger(\tau)B_\beta \rangle$ is the environment correlation function (the average $\langle \cdot \rangle$ is calculated on the stationary state of the environment). This is one of the many forms of the so-called Redfield equation [8,9], and it is obtained under the assumption that the typical evolution time $\tau_S$ of $\widetilde{\rho}(t)$ is the longest timescale of the problem (see [20] for further discussions).

Our first task is to write the Redfield equation in the form (2). Let us consider the basis $\{|k\rangle\}$ of normalized eigenvectors of the free system Hamiltonian $H_S$, so that we can write its spectral decomposition as $H_S = \sum_k \omega_k E_{kk}$, where $E_{kq} := |k\rangle\langle q|$. Since $\{E_{kq}\}_{k,q=1,\dots,N}$ is an orthonormal basis of $\mathrm{L}(\mathcal{H}_S)$ we can expand, for example,

$$A_\beta = \sum_{k,q} A_{\beta,kq} E_{kq} \,, \tag{6}$$

where $A_{\beta,kq} = \langle k|A_\beta|q\rangle$. We have now to replace the decomposition (6) inside Eq. (5): the details are reported in App. A. Going back to the Schrödinger picture, one finds

$$\frac{d\rho(t)}{dt} = -i[H_S + H_{LS}(t), \rho(t)] + \sum_{k,q,n,m} \chi_{kq,nm}(t)\left[E_{kq}\rho(t)E_{nm}^\dagger - \frac{1}{2}\{E_{nm}^\dagger E_{kq}, \rho(t)\}\right], \tag{7}$$

where

$$\chi_{kq,nm}(t) = \sum_{\alpha,\beta}\left[\Gamma_{\alpha\beta}(\omega_{kq}, t) + \Gamma_{\beta\alpha}^*(\omega_{nm}, t)\right]A_{\beta,kq}A_{\alpha,nm}^* \,, \tag{8}$$

$$H_{LS}(t) = \sum_{k,q,n,m} \eta_{kq,nm}(t) E_{nm}^\dagger E_{kq} \,, \quad \eta_{kq,nm}(t) = \sum_{\alpha,\beta} \frac{\Gamma_{\alpha\beta}(\omega_{kq}, t) - \Gamma_{\beta\alpha}^*(\omega_{nm}, t)}{2i} A_{\beta,kq}A_{\alpha,nm}^* \,, \tag{9}$$

and

$$\Gamma_{\alpha\beta}(\omega, t) := \int_0^t d\tau\, c_{\alpha\beta}(\tau)e^{i\omega\tau}\,. \tag{10}$$

We also introduced the Bohr frequencies $\omega_{kq} := \omega_q - \omega_k$. Since $\chi$ and $H_{LS}$ generally depend on time, we do not have an equation of the form (2) yet. A common way to strictly obtain an equation in the form (2) is to perform the second Markov approximation [6], which consists in replacing $\int_0^t \to \int_0^\infty$ in (10). In this scenario, we will write $\Gamma_{\alpha\beta}(\omega) = \lim_{t\to\infty}\Gamma_{\alpha\beta}(\omega, t)$. For future reference, notice that if we split $\Gamma_{\alpha\beta} = J_{\alpha\beta} + iS_{\alpha\beta}$ in its real and imaginary part, we have

$$J_{\alpha\beta}(\omega) = \frac{1}{2}\int_{-\infty}^{\infty} d\tau\, c_{\alpha\beta}(\tau)e^{i\omega\tau} = \frac{\hat{c}_{\alpha\beta}(\omega)}{2}\,, \tag{11}$$

where $\hat{c}_{\alpha\beta}$ is the Fourier transform of $c_{\alpha\beta}$. Therefore one can invoke Bochner's theorem to infer that $J$ is a positive semidefinite matrix [6].

In the general time-dependent case it is difficult to say when the Redfield equation leads to CPT dynamics, since (to our knowledge) there have not yet been found necessary conditions for complete positivity of a time-dependent generator. However, we do know a sufficient condition, namely $\chi(t) \geq 0$ for all $t \geq 0$. Unfortunately (as we shall also see in Sec. 3.2), $\chi(t)$ is not positive semidefinite in general for the Redfield equation. In the time-independent case, this is a long-standing well-known problem [10].

# 3 Regularization procedures

## 3.1 Common existing regularizations

We argue that most of the procedures to recover positivity from the Redfield equation are in fact ways to transform the matrix $\chi$ in Eq. (8) into a positive semidefinite one, and the vast majority of them only deals with the time-independent case. For example, a popular approach consists in performing a coarse-graining transformation in the equation for $\widetilde{\rho}(t)$ [18]. In our notations, this means that we apply the operation

$$\mathcal{C}_{\Delta t}(X(t)) := \frac{1}{\Delta t} \int_{t-\Delta t/2}^{t+\Delta t/2} X(s)ds, \tag{12}$$

to both sides of Eq. (A.2). The advantage lies in the fact that if we choose $\Delta t \ll \tau_{\mathcal{S}}$, where $\tau_{\mathcal{S}}$ is the timescale of variation of $\widetilde{\rho}(t)$, we can ignore the action of $\mathcal{C}_{\Delta t}$ on $\widetilde{\rho}(t)$ and pull the latter out of the integral. Using the fact that

$$\frac{1}{\Delta t} \int_{t-\Delta t/2}^{t+\Delta t/2} e^{i(\omega_{nm}-\omega_{kq})s}ds = \text{sinc}\left(\frac{(\omega_{nm}-\omega_{kq})\Delta t}{2}\right), \tag{13}$$

where $\text{sinc}(x) = \sin(x)/x$ is the cardinal sinus, we see that the effect of the coarse graining is given by the substitution

$$\chi_{kq,nm} \to \chi_{kq,nm}^{(\Delta t)} := \chi_{kq,nm}\text{sinc}\left(\frac{(\omega_{nm}-\omega_{kq})\Delta t}{2}\right), \tag{14}$$

and similarly for the Lamb shift coefficient $\eta_{kq,nm} \to \eta_{kq,nm}^{(\Delta t)}$. The interesting fact about this expression is that one can prove that if $\Delta t$ is sufficiently high the matrix $\chi^{(\Delta t)}$ will be positive semidefinite [18]. In the extreme situation $\Delta t \to \infty$ one has

$$\chi_{kq,nm}^{(\infty)} = \chi_{kq,nm}\delta_{\omega_{kq},\omega_{nm}}, \tag{15}$$

which is the Kossakowski matrix obtained with a secular approximation [6]. For this reason one also says that $\chi^{(\Delta t)}$ with general (but appropriate) $\Delta t$ is the Kossakowski matrix in *partial secular approximation*.

A more recent and permissive construction is the one provided by Nathan and Rudner in Ref. [22] and by Davidović in Ref. [5]. The idea is to replace the arithmetic mean that appears in (8) with a geometric one:

$$\Gamma_{\alpha\beta}(\omega_{kq}) + \Gamma_{\beta\alpha}^{*}(\omega_{nm}) \to 2\left[\sqrt{J(\omega_{nm})}\sqrt{J(\omega_{kq})}\right]_{\alpha\beta}, \tag{16}$$

where it is intended matrix multiplication of matrix square roots (recall that $J \geq 0$). It can be shown that this approach is justified whenever $\tau_{\mathcal{S}} \gg 1/\omega_R$, where $\omega_R$ is representative of the frequency range of the system [34].

## 3.2 New regularization

Since we have to regularize the Kossakowski matrix, the following proposal seems very natural: for every $t \geq 0$, replace $\chi(t)$ with its closest positive semidefinite matrix of the same dimension. More precisely, given a norm $\|\cdot\|$ on the space of $N \times N$ complex matrices we define

$$\chi^+(t) := \arg \min_{P = P^\dagger \geq 0} \|\chi(t) - P\|, \tag{17}$$

and use $\chi^+(t)$ instead of $\chi(t)$ in the Redfield equation, thus obtaining a LGKS-like equation. Unlike standard approaches, notice that here we are retaining the time dependence of $\chi$. If we choose the Frobenius norm $\|X\|_F = \sqrt{\text{Tr}[X^\dagger X]}$, an explicit formula for $\chi^+(t)$ exists [35]. Since in our case $\chi(t)$ is Hermitian, Ref. [35] shows that the same expression is obtained by using the spectral norm $\|X\|_\infty = \sigma_{\max}(X)$, where $\sigma_{\max}(X)$ is the maximum singular value of $X$. The result is that $\chi^+(t)$ is the "positive part" of $\chi(t)$, obtained from $\chi(t)$ by putting to zero the negative eigenvalues:

$$\chi^+(t) = \frac{\chi(t) + \sqrt{\chi^\dagger(t)\chi(t)}}{2}. \tag{18}$$

This gives a fairly general efficient way to determine $\chi^+(t)$, at least numerically: it is sufficient to compute a spectral decomposition.

In order to gain some understanding we will now make some observations about the spectral structure of $\chi(t)$ in (8). For notational convenience, here we will not write the time parameter and we will use collective indices $i = (k, q)$ and $j = (n, m)$, lexicographically ordered. Moreover we write $\Gamma_{\alpha\beta,i}$ to mean $\Gamma_{\alpha\beta}(\omega_{kq}, t)$. Then we have

$$\chi_{ij} = \sum_{\alpha,\beta} \left( \Gamma_{\alpha\beta,i} + \Gamma^*_{\beta\alpha,j} \right) A_{\beta,i} A^*_{\alpha,j} = \sum_\alpha \left( G_{\alpha,i} A^*_{\alpha,j} + A_{\alpha,i} G^*_{\alpha,j} \right), \tag{19}$$

where $G_{\alpha,i} := \sum_\beta \Gamma_{\alpha\beta,i} A_{\beta,i}$. If we define the vectors $|A_\alpha\rangle = \sum_i A_{\alpha,i} |i\rangle$ and $|G\rangle_\alpha = \sum_i G_{\alpha,i} |i\rangle$ one can easily verify that

$$\chi = \sum_\alpha (|A_\alpha\rangle\langle G_\alpha| + |G_\alpha\rangle\langle A_\alpha|). \tag{20}$$

Up to now $|A_\alpha\rangle$ and $|G_\alpha\rangle$ are general vectors. Since every Hermitian matrix can be written in the form (20), it is quite difficult to say something general about its spectrum.

A case that can be treated explicitly is when there is only one noise channel $M = 1$. Here we can drop the $\alpha, \beta$ indices and obtain

$$\chi = |A\rangle\langle G| + |G\rangle\langle A|. \tag{21}$$

Let us ignore the trivial cases in which $A_i \equiv 0$ or $\Gamma_i \equiv 0$, which would lead to $\chi = 0$. Then it is easy to see that the vectors $|A\rangle$ and $|G\rangle$ are linearly independent, unless $\Gamma_i \equiv \Gamma \neq 0$, in which case $|G\rangle = \Gamma |A\rangle$. In the latter scenario $\chi = 2 \text{Re}\,\Gamma |A\rangle\langle A|$: this is a rank-one matrix with nonzero eigenvalue $\lambda = 2 \text{Re}\,\Gamma \|A\|^2$ and associated normalized eigenvector $|A\rangle / \|A\|$, where $\|A\|^2 = \sum_i |A_i|^2$. This case is not so interesting because, at least when $t \to \infty$, $\lambda \geq 0$ by Bochner's theorem and no regularization is needed. However note that many common models based on qubits and harmonic oscillators resonantly coupled with a bosonic bath fall exactly in the case mentioned above (see App. B for details). Suppose instead that $|A\rangle$ and $|G\rangle$ are independent. Then $\chi$ is a rank-two matrix and the eigenvectors associated with nonzero eigenvalues are of the form $|v\rangle = a |G\rangle + b |A\rangle$. Writing $\chi |v\rangle = \lambda |v\rangle$ and equating coefficients we find two solutions:

$$\lambda_\pm = \text{Re}\,\langle G|A\rangle \pm \sqrt{\|G\|^2 \|A\|^2 - \text{Im}^2 \langle G|A\rangle}, \tag{22a}$$

$$\left(\frac{a}{b}\right)_\pm = \frac{\lambda_\pm - \langle G|A\rangle}{\|G\|^2}. \tag{22b}$$

Notice that by the Cauchy-Schwarz inequality $\lambda_+ \geq 0$ and $\lambda_- \leq 0$, and we confirm that $\chi$ is not positive semidefinite in general [10], even in the time-dependent case.

It is instructive to rewrite these expressions in terms of physical quantities, which are encoded in $\Gamma$. Given a vector $x \in \mathbb{R}^{N^2}$ let us define

$$\langle x \rangle := \frac{\sum_i x_i |A_i|^2}{\sum_i |A_i|^2}, \tag{23}$$

a notation that treats $|A_i|^2$ as a probability distribution. Then

$$\lambda_\pm = \|A\|^2 [\langle J \rangle \pm \mathcal{V}(\Gamma)], \tag{24}$$

where we defined for convenience the quantity

$$\mathcal{V}(\Gamma) := \sqrt{\langle J \rangle^2 + \text{Var}(J) + \text{Var}(S)}, \tag{25}$$

and $\text{Var}(x) := \langle x^2 \rangle - \langle x \rangle^2$ is the variance of $x \in \mathbb{R}^{N^2}$.

A similar result was obtained in Refs. [15, 25, 34]. In particular, in Ref. [25] the authors parametrically splitted the Redfield equation in a "positive" and a "negative" contribution, minimizing the latter with an optimized choice of the parameters. For a single noise channel this is equivalent to our formulation, since if we write $\chi = \chi^+ + \chi^-$, where $\chi^- := (\chi - \sqrt{\chi^\dagger \chi})/2$, then $\|\chi - \chi^+\| = \|\chi^-\|$ and we know that $\chi^+$ minimizes $\|\chi - P\|$ for positive semidefinite $P$. Our approach generalizes this view, since it provides a well-defined procedure to regularize the Kossakowski matrix with an arbitrary number of (even correlated) noise channels.

Notice that the bigger the variances in Eq. (25) the bigger the magnitude of the negative eigenvalue $\lambda_-$, hence the regularization is expected to cause minimum disturbance when the environment correlation function is quite flat over the set of Bohr frequencies of the system. This is consistent with a Markovian dynamics requirement. In fact, the magnitude of the negative eigenvalue of the Kossakowski matrix has been used before to quantify non-Markovianity [36], and is related to the well-known non-Markovianity measure introduced by Rivas, Huelga, and Plenio in Ref. [37].

To conclude, let us provide the expression for the regularized Kossakowski matrix in the single noise channel scenario [cf. Eq. (21)]. It is given by $\chi^+ = \lambda_+ |+\rangle\langle+|$, where $|+\rangle$ is the normalized eigenvector associated with $\lambda_+$. For simplicity, let us indicate here $\lambda \equiv \lambda_+$. Given the shape of $a/b$ in (22), consider the eigenvector

$$|v\rangle = (\lambda - \langle G|A \rangle) |G\rangle + \|G\|^2 |A\rangle . \tag{26}$$

A calculation shows that

$$\|v\|^2 = 2\lambda \|G\|^2 \|A\|^2 \mathcal{V}(\Gamma). \tag{27}$$

The normalized eigenvector is therefore $|+\rangle = a |G\rangle + b |A\rangle$ with

$$a = \frac{\lambda - \langle G|A \rangle}{\|v\|}, \qquad b = \frac{\|G\|^2}{\|v\|}, \tag{28}$$

and then $\chi^+ = \lambda(a |G\rangle + b |A\rangle)(a^* \langle G| + b^* \langle A|)$. Using the expressions given above, one finds after some algebra that the components of $\chi^+$ are

$$\chi_{ij}^+ = \frac{A_i A_j^*}{2\mathcal{V}(\Gamma)} \Big[ \Gamma_i \Gamma_j^* + \langle J^2 \rangle + \langle S^2 \rangle + (\mathcal{V}(\Gamma) + i\langle S \rangle)\Gamma_i + (\mathcal{V}(\Gamma) - i\langle S \rangle)\Gamma_j^* \Big]. \tag{29}$$

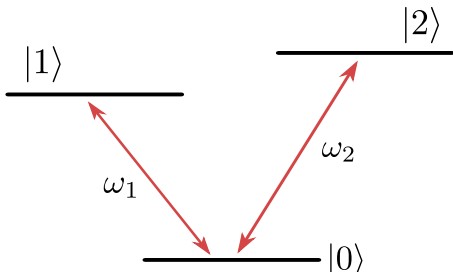

Figure 1: Open three-level V-system described in Sec. 4. The arrows indicate transitions induced by an external bosonic environment in its vacuum state.

## 4 Example: Open three-level system

Now we want to compare different regularization procedures for a system that can be solved exactly, in order to try to assess which performs better. We will study a minimal model that is sufficiently complex to show the feasibility of our approach. A well-known example of an exactly solvable open quantum system is the spontaneous decay of a qubit into the field vacuum [6]. As it will be clear later, this setting is not complex enough for our purposes since the Kossakowski matrix turns out to be rank-one positive in the limit $t \to \infty$. A less known fact is that some kinds of open three-level systems can also be solved exactly [38], so we take that route.

Consider the V-system depicted in Fig. 1, with free Hamiltonian

$$H_{\mathcal{S}} = \omega_1 |1\rangle\langle 1| + \omega_2 |2\rangle\langle 2|, \qquad \omega_{1,2} > 0 \tag{30}$$

(we assume the ground state $|0\rangle$ to be at zero energy). The environment is an infinite collection of bosonic modes with Hamiltonian $H_{\mathcal{E}} = \sum_p \epsilon_p b_p^\dagger b_p$ and placed in the respective vacuum $|\Omega\rangle$. The interaction is a linear coupling that causes transitions between the levels $|0\rangle$ and $|1\rangle$ and between the levels $|0\rangle$ and $|2\rangle$:

$$H_I = \sum_p \left( g_{1,p} |0\rangle\langle 1| + g_{2,p} |0\rangle\langle 2| \right) \otimes b_p^\dagger + \text{H.c.}, \tag{31}$$

where the coupling constants $g_{1,p}$ and $g_{2,p}$ are assumed for simplicity to be real numbers.

For the purpose of writing the Redfield equation, a simple calculation shows that the only relevant coupling operators are

$$A_\alpha = |0\rangle\langle \alpha|, \quad B_\alpha = \sum_p g_{\alpha,p} b_p^\dagger, \qquad \alpha \in \{1, 2\}. \tag{32}$$

The others will make $c_{\alpha\beta} = 0$ and hence do not appear in the final master equation. Instead, for these we have $c_{\alpha\beta}(\tau) = \sum_p g_{\alpha,p} g_{\beta,p} e^{-i\epsilon_p \tau} \neq 0$. For example, let us assume

$$c_{\alpha\beta}(\tau) = \frac{\gamma_{\alpha\beta}\mu}{2} e^{-\mu|\tau|} e^{-i\omega_0 \tau}, \tag{33}$$

where $\mu, \omega_0 > 0$ and $\gamma_{\alpha\beta} = \sqrt{\gamma_\alpha \gamma_\beta}$ with $\gamma_1, \gamma_2 > 0$. This exponential shape comes from a Lorentzian bath assumption with

$$J_{\alpha\beta}(\omega) = \frac{\gamma_{\alpha\beta}\mu}{2} \frac{\mu}{(\omega - \omega_0)^2 + \mu^2}. \tag{34}$$

This is the choice that was made in Ref. [38] and we follow it here to provide a direct comparison between the exact dynamics and the various master equations. In order to make the

paper self-contained we provide in App. C the derivation of the exact solution that is used in the following numerical calculations [cf. Eq. (C.3) and Eq. (C.12)].

In the simplifying case $\gamma_1 = \gamma_2 = \gamma$, we can consider $\gamma$ as an estimate of the inverse evolution time of the system: $\gamma \sim 1/\tau_S$. Moreover, we can take $\mu$ as an estimate of the inverse decay time of environment's correlations: $\mu \sim 1/\tau_{\mathcal{E}}$. As a consequence, the Markovian approximation consists in assuming $\gamma \ll \mu$.

## 4.1 Numerical comparison

The structure of the Redfield equation [cf. Eq. (8)] is determined by the presence of the factor

$$A_{\beta,kq}A^*_{\alpha,nm} = \langle k|0\rangle\langle\beta|q\rangle\langle 0|n\rangle\langle m|\alpha\rangle. \tag{35}$$

This means that the only nonzero entries of $\chi(t)$ occur for $k = n = 0$ and $q = \beta$, $m = \alpha$. With a quick calculation one realizes that

$$\frac{d\rho(t)}{dt} = -i[H_S + H_{LS}(t), \rho(t)] + \sum_{\alpha,\beta} d_{\alpha\beta}(t)\left(\langle\beta|\rho(t)|\alpha\rangle|0\rangle\langle 0| - \frac{1}{2}\{|\alpha\rangle\langle\beta|, \rho(t)\}\right), \tag{36}$$

where

$$d_{\alpha\beta}(t) := \Gamma_{\alpha\beta}(\omega_\beta, t) + \Gamma^*_{\beta\alpha}(\omega_\alpha, t), \tag{37}$$

and the Lamb shift is $H_{LS}(t) = \sum_{\alpha,\beta} h_{\alpha\beta}(t)|\alpha\rangle\langle\beta|$ with

$$h_{\alpha\beta}(t) := \frac{1}{2i}\left[\Gamma_{\alpha\beta}(\omega_\beta, t) - \Gamma^*_{\beta\alpha}(\omega_\alpha, t)\right]. \tag{38}$$

We can also conveniently rewrite

$$\frac{d\rho(t)}{dt} = \sum_{\alpha,\beta}\left[d_{\alpha\beta}(t)\rho_{\beta\alpha}(t)|0\rangle\langle 0| + \phi_{\alpha\beta}(t)\rho(t)|\alpha\rangle\langle\beta| + \phi^*_{\beta\alpha}(t)|\alpha\rangle\langle\beta|\rho(t)\right], \tag{39}$$

where $\rho_{\beta\alpha}(t) = \langle\beta|\rho(t)|\alpha\rangle$ and

$$\phi_{\alpha\beta}(t) := i\delta_{\alpha\beta}\omega_\alpha + ih_{\alpha\beta}(t) - \frac{1}{2}d_{\alpha\beta}(t). \tag{40}$$

With respect to the basis $\{|0\rangle, |1\rangle, |2\rangle\}$ this can also be written in components as

$$\begin{aligned}
\dot{\rho}_{00} &= d_{11}\rho_{11} + d_{21}\rho_{12} + d_{12}\rho_{21} + d_{22}\rho_{22}, \\
\dot{\rho}_{01} &= \phi_{11}\rho_{01} + \phi_{21}\rho_{02}, \\
\dot{\rho}_{02} &= \phi_{12}\rho_{01} + \phi_{22}\rho_{02}, \\
\dot{\rho}_{11} &= -d_{11}\rho_{11} + \phi_{21}\rho_{12} + \phi^*_{21}\rho_{21}, \\
\dot{\rho}_{12} &= \phi_{12}\rho_{11} + (\phi^*_{11} + \phi_{22})\rho_{12} + \phi^*_{21}\rho_{22}, \\
\dot{\rho}_{22} &= \phi^*_{12}\rho_{12} + \phi_{12}\rho_{21} - d_{22}\rho_{22},
\end{aligned} \tag{41}$$

where we dropped the time dependence for notational convenience. This is a linear system of first-order differential equations that can be efficiently solved by a numerical routine: here we adopted a Runge-Kutta method (RK45) provided by the Python library SciPy [39, 40].

Notice that the Kossakowski matrix for this setting (which is $9 \times 9$) is filled with zeros except for a $2 \times 2$ block with entries $d_{\alpha\beta}$. Therefore it is clear that regularizing $\chi$ is equivalent to regularizing $d$. By choosing a two-level system instead of a three-level one the nonzero block would have consisted of a single entry: this scenario would be trivial since positivity is

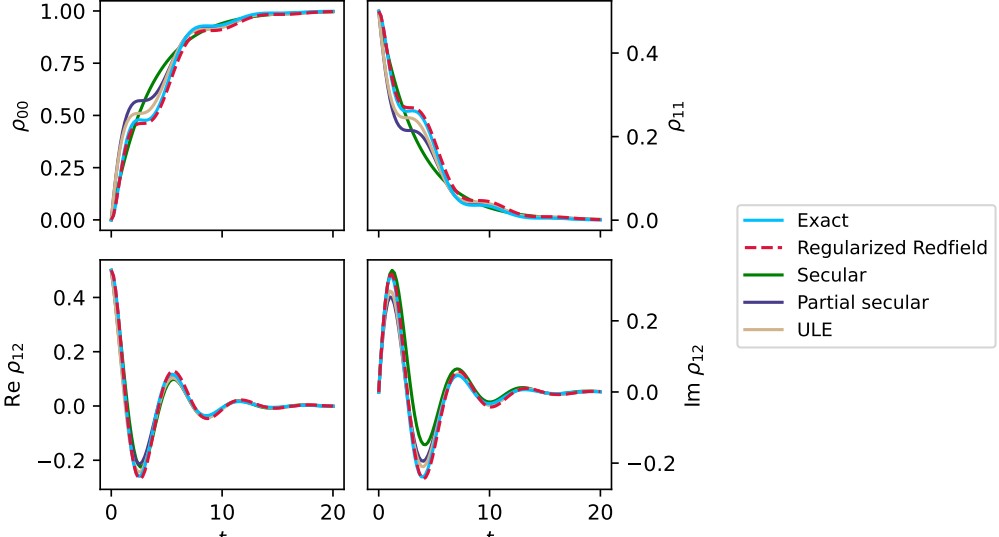

Figure 2: Time evolution of the three-level density matrix elements starting from the pure state $|\psi_0\rangle = (|1\rangle + |2\rangle)/\sqrt{2}$. The label "Regularized Redfield" refers to the proposal of the present paper. The partial secular case [cf. (14)] is obtained by finding the smallest coarse-graining time that guarantees positivity of the Kossakowski matrix. The label "ULE" refers to the "universal Lindblad equation" described in Ref. [22] [also, cf. (16)]. Here $\omega_1 = 1$, $\omega_2 = 2$, $\omega_0 = 1.5$, $\gamma_1 = \gamma_2 = 0.3$, and $\mu = 2$.

then guaranteed by Bochner's theorem, at least in the time-independent case. See App. B for clarifications on this point.

Now we present a comparison between the exact solution provided by Ref. [38] [reported here in App. C, cf. Eq. (C.3) and Eq. (C.12)] and what we obtain by numerically solving the system in (41) for various choices of regularization of the matrix $d$. In Fig. 2 we report the results for the evolution starting from the pure initial state $|\psi_0\rangle = (|1\rangle + |2\rangle)/\sqrt{2}$, and choosing as parameters $\omega_1 = 1$, $\omega_2 = 2$, $\omega_0 = 1.5$, $\gamma_1 = \gamma_2 = 0.3$, and $\mu = 2$. At this level all equations behave more or less similarly. Except for the fact that the secular-approximated one globally provides the worst results, it is hard to tell which of the others performs better. The situation is similar for other choices of parameters.

## 4.2 Choi operator technique

If we want to assess more carefully the quality of a regularization procedure we should find a way to compare the results with the exact one in a way that is independent from the initial state. In order to do that, let us step back to the dynamical semigroup picture of Sec. 2. What we actually want is to compare the semigroup $\{\Phi_{t,s}\}$ generated by our master equation with the semigroup $\{\Phi_{t,s}^{(e)}\}$ generated by the exact dynamics. Here we propose a simple approach to compare them pointwise, i.e., at fixed time $t$. More global comparisons should be possible but are out of the scope of the present paper and are left to future work.

Given a map $\Phi_{t,s} : \mathrm{L}(\mathcal{H}_{\mathcal{S}}) \to \mathrm{L}(\mathcal{H}_{\mathcal{S}})$ we can construct the Choi operator [27,28]

$$\mathcal{J}(\Phi_{t,s}) := \sum_{n,m=1}^{N} \Phi_{t,s}(E_{nm}) \otimes E_{nm} \in \mathrm{L}(\mathcal{H}_{\mathcal{S}} \otimes \mathcal{H}_{\mathcal{S}}). \tag{42}$$

A well-known fact is that $\Phi_{t,s}$ is completely positive if and only if $\mathcal{J}(\Phi_{t,s}) \geq 0$. However, we are mostly interested in the fact that there exists a bijection $\Phi_{t,s} \longleftrightarrow \mathcal{J}(\Phi_{t,s})$, which is the

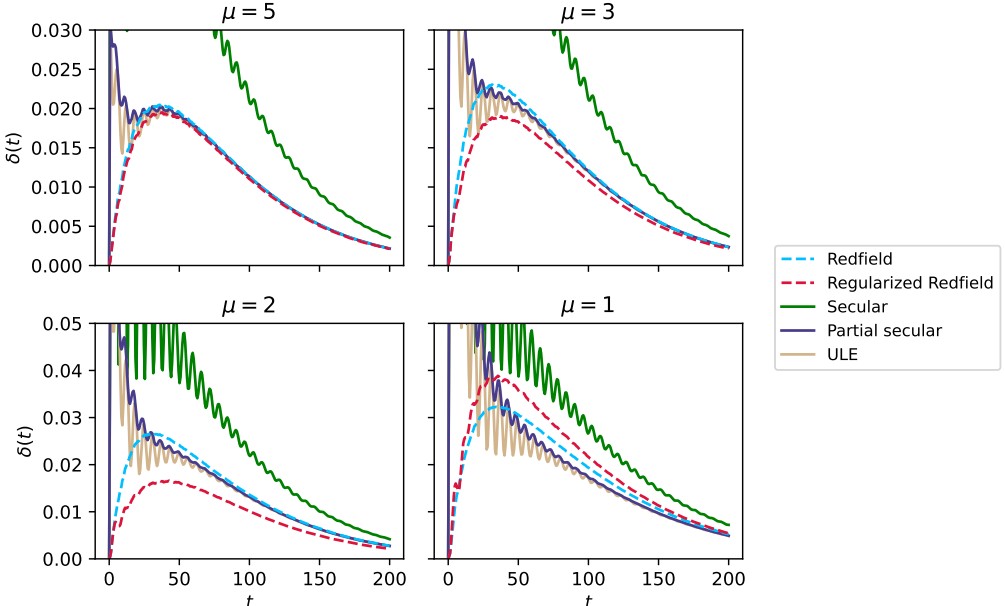

Figure 3: Distance from the exact dynamics for the three-level system as represented by the quantity $\delta(t)$, defined in Eq. (43) using the Frobenius norm, for several values of $\mu$ while fixing $\omega_1 = 1$, $\omega_2 = 2$, $\omega_0 = 1.5$, $\gamma_1 = \gamma_2 = 0.05$. The various master equations are chosen as in Fig. 2.

Choi-Jamiołkowski isomorphism. The usefulness of this observation is twofold. First of all, on $L(\mathcal{H}_S \otimes \mathcal{H}_S)$ we have well-established metrics that we can use, such as the Frobenius norm. Secondly, if we compute

$$\delta(t) := \|\mathcal{J}(\Phi_{t,0}) - \mathcal{J}(\Phi_{t,0}^{(e)})\| \,, \tag{43}$$

we have a (pointwise) measure of the difference between the two dynamics that does not depend on the initial state.

In Fig. 3 we report examples for $\delta(t)$ calculated with the Frobenius norm for various master equations. Here we fix $\omega_1 = 1$, $\omega_2 = 2$, $\omega_0 = 1.5$, $\gamma_1 = \gamma_2 = 0.05$, and we present plots for several values of the spectral width $\mu$. We find that the performance of our approach with respect to the others depends on the ratio between $\mu$ and the frequency range of the system $\omega_R = \max\{\omega_1, \omega_2\}$.

For values of $\mu$ sufficiently higher than $\omega_R$ our procedure has little effect on the already good accuracy of the Redfield equation, as expected from the fact that the Kossakowski matrix is essentially positive in a deep Markovian regime (see discussion in Sec. 3.2). For smaller values of $\mu$ (while still being greater than $\omega_R$) we can instead observe how our version of the regularized Redfield equation approximates well the exact dynamics in a more consistent way with respect to the other master equations, especially at short times. This was not clear with a direct comparison at the level of the density matrix, and it is a consequence of our ability to retain time dependence in the Kossakowski matrix.

However, a change in the trend can be observed when lowering the value of $\mu$ below $\omega_R$, where our regularization provides results worse than Redfield itself. In this essentially non-Markovian regime the truncation of the negative part of the Kossakowski matrix has a too drastic effect on the dynamics. This situation is of course out of reach for the approach of the present paper, since it enforces CP-divisible dynamics, and other methods should be used.

# 5 Conclusions

In this work we looked at the problem of finding a LGKS-like equation from the microscopic dynamics as a regularization process of the Kossakowski matrix in the Redfield equation. With this picture in mind, we proposed to replace such a matrix with its closest positive semidefinite one, thus providing the CP-divisible dynamics that is closest to the Redfield one. We also used the Choi-Jamiołkowski isomorphism to envision a pointwise measure of the distance between two dynamical processes, and we applied it to the problem of assessing which master equation better approximates the exact dynamics of a simple open three-level system. We found our proposal to lead to the overall best results in this regard, provided one works in a Markovian regime where the spectral width of the environment is greater than the frequency range of the system. Notably, our approach is tailored to retain the time dependence of the Kossakowski matrix, allowing it to be accurate at short times. Unfortunately, at this level the approach is mainly numerical and it is still an open problem to understand what are the implications of the proposed manipulation on the thermodynamics of the system and the steady-state manifold structure. Note that at the Redfield level these kinds of characterizations already present some subtleties: for a system in contact with a finite-temperature bath the steady state is not the Gibbs state of the system and corrections should be included by considering a mean force Gibbs state [41]. Such calculations depend on the shape of the Kossakowski matrix, which we are modifying in an analytically unpredictable way (at least in the general case), thus making an immediate transition to the regularized scenario difficult.

A possible future improvement would be to envision an alternative regularization scheme that is able to retain the non-Markovian features of the Redfield equation. Moreover, it would be desirable to have a meaningful measure of the distance between two dynamical processes which goes beyond the pointwise approach followed here: this is an interesting problem on its own and can lead to other general applications.

Another question that needs to be addressed is to what extent our conclusions can be applied to infinite-dimensional systems, where it is trickier to apply the LGKS theorem and where we expect the choice of the involved norms to matter more.

## Acknowledgments

V.G. and A.D. acknowledge financial support by MIUR (Ministero dell'Istruzione, dell'Università e della Ricerca) by PRIN 2017 "Taming complexity via QUantum Strategies: a Hybrid Integrated Photonic approach" (QUSHIP) Id. 2017SRN-BRK, and by project PRO3 "Quantum Pathfinder". V.C. is supported by the Luxembourg national research fund in the frame of Project QUTHERM C18/MS/12704391.

# A Standard form of the Redfield equation

In this appendix we see how to write the Redfield equation (5) resulting from the Born-Markov approximation, in the form (2). This step is necessary to derive the Kossakowski matrix associated with the Redfield equation, check that is not positive semidefinite in general and discuss eventual regularization procedures. In the interaction picture the decomposition (6) becomes

$$\widetilde{A}_\beta(t-\tau) = \sum_{k,q} A_{\beta,kq} e^{-i\omega_{kq}(t-\tau)} E_{kq}, \qquad (A.1)$$

where we introduced the Bohr frequencies $\omega_{kq} := \omega_q - \omega_k$ associated with the jumps between the eigenstates of the system Hamiltonian $|q\rangle$ and $|k\rangle$. Applying Eq. (A.1) to $\widetilde{A}_\beta(t-\tau)$ and $\widetilde{A}_\alpha^\dagger(t)$ in Eq. (5) we end up with

$$\frac{d\widetilde{\rho}(t)}{dt} = \sum_{\alpha,\beta} \sum_{k,q,n,m} \Gamma_{\alpha\beta}(\omega_{kq},t) A_{\beta,kq} A_{\alpha,nm}^* e^{i(\omega_{nm}-\omega_{kq})t} [E_{kq}\widetilde{\rho}(t), E_{nm}^\dagger] + \text{H.c.}, \tag{A.2}$$

where we introduced the quantity $\Gamma_{\alpha\beta}$ defined in Eq. (10) of the main text. The exponential factor in (A.2) can be eliminated by going back to the Schrödinger picture:

$$\frac{d\rho(t)}{dt} = -i[H_\mathcal{S}, \rho(t)] + \sum_{k,q,n,m} \left( K_{kq,nm}(t)[E_{kq}\rho(t), E_{nm}^\dagger] + \text{H.c.} \right), \tag{A.3}$$

where we defined the matrix

$$K_{kq,nm}(t) := \sum_{\alpha,\beta} \Gamma_{\alpha\beta}(\omega_{kq},t) A_{\beta,kq} A_{\alpha,nm}^* . \tag{A.4}$$

Let us focus on the term inside the round brackets in Eq. (A.3). If we expand the commutator and write explicitly the "H.c." part we get

$$K_{kq,nm}[E_{kq}\rho, E_{nm}^\dagger] + \text{H.c.} = K_{kq,nm}(E_{kq}\rho E_{nm}^\dagger - E_{nm}^\dagger E_{kq}\rho) + K_{kq,nm}^*(E_{nm}\rho E_{kq}^\dagger - \rho E_{kq}^\dagger E_{nm}), \tag{A.5}$$

where we dropped the time dependence of $\rho(t)$ and $K_{kq,nm}(t)$ for ease of notation. It is convenient to treat the first and third term of the right-hand side together. Replacing them in the sum in Eq. (A.3) we have

$$\sum_{k,q,n,m} (K_{kq,nm}E_{kq}\rho E_{nm}^\dagger + K_{kq,nm}^* E_{nm}\rho E_{kq}^\dagger) = \sum_{k,q,n,m} (K_{kq,nm} + K_{nm,kq}^*)E_{kq}\rho E_{nm}^\dagger . \tag{A.6}$$

The second and fourth term of Eq. (A.5) can be treated similarly:

$$\sum_{k,q,n,m} (K_{kq,nm}E_{nm}^\dagger E_{kq}\rho + K_{kq,nm}^*\rho E_{kq}^\dagger E_{nm}) = \sum_{k,q,n,m} (K_{kq,nm}E_{nm}^\dagger E_{kq}\rho + K_{nm,kq}^*\rho E_{nm}^\dagger E_{kq})$$

$$= \frac{1}{2}\sum_{k,q,n,m} \left\{ (K_{kq,nm} + K_{nm,kq}^*)\{E_{nm}^\dagger E_{kq}, \rho\} + (K_{kq,nm} - K_{nm,kq}^*)[E_{nm}^\dagger E_{kq}, \rho] \right\}. \tag{A.7}$$

Equation (7) is obtained by plugging Eqs. (A.6) and (A.7) in Eq. (A.3) and introducing the matrices $\eta_{kq,nm}$, $\chi_{kq,nm}$ defined in Sec. 2 of the main text.

# B  Kossakowski matrix for a qubit and a harmonic oscillator

In this section we compute the Kossakowski matrix of two common scenarios, in which a bosonic bath is coupled with either a qubit or a harmonic oscillator. Although simple, these examples share an interesting peculiarity: the Kossakowski matrix of the corresponding time-independent Redfield equation is positive semidefinite, so that no regularization is needed to ensure the positivity of the dynamics.

In the first model a qubit is coupled to a bath of harmonic oscillators with a rotating-wave interaction Hamiltonian. Denoting the two energy levels of the qubit as $|0\rangle, |1\rangle$ we have $H_\mathcal{S} = \omega_1 |1\rangle\langle 1|$, $H_\mathcal{E} = \sum_p \epsilon_p b_p^\dagger b_p$, while the interaction Hamiltonian writes

$$H_I = \sum_p g_{1,p} |0\rangle\langle 1| \otimes b_p^\dagger + \text{H.c.}, \tag{B.1}$$

where $b_p$ is the creation operator relative to the environmental mode $p$. Using the notation of Eq. (4) we have $A_1 = |0\rangle\langle 1|$, and $B_1 = \sum_p g_{1,p} b_p^\dagger$, while the Hermitian conjugates of $A_1, B_1$ do not contribute to the dynamical equation since the relative correlation function $\langle \check{b}_p^\dagger(\tau) b_p \rangle$ vanishes. The coordinates of $A_1$ in the decomposition (6) are simply given by $A_{1,kq} = \delta_{k,0}\delta_{q,1}$ so that

$$K_{kq,nm}(t) = \Gamma_{11}(\omega_{kq}, t)\delta_{k,0}\delta_{q,1}\delta_{n,0}\delta_{m,1} = \Gamma_{11}(\omega_{01}, t)\delta_{k,0}\delta_{q,1}\delta_{n,0}\delta_{m,1}. \tag{B.2}$$

Then the Kossakowski matrix reads

$$\chi_{kq,nm}(t) = [\Gamma_{11}(\omega_1, t) + \Gamma_{11}^*(\omega_1, t)]\delta_{k,0}\delta_{q,1}\delta_{n,0}\delta_{m,1}, \tag{B.3}$$

where we used $\omega_{01} = \omega_1$. The matrix in Eq. (B.3) is diagonal, with the single non-zero element being $\chi_{01,01}(t)$. After the second Markov approximation is applied, we are left with $\chi_{01,01} = 2\operatorname{Re}\Gamma_{11}(\omega_1)$ that is ensured to be positive as a consequence of Bochner's theorem.

The case of the harmonic oscillator can be treated similarly. While the bath Hamiltonian is the same as the preceding example, we have $H_S = \omega_S a^\dagger a$ and

$$H_I = \sum_p g_{1,p} a \otimes b_p^\dagger + \text{H.c.}, \tag{B.4}$$

where $a, a^\dagger$ are creation and annihilation operators of the system. We repeat the calculations done for the qubit, but considering $A_1 = a$ and obtaining $A_{1,kq} = \sum_{l_1=0}^\infty \sqrt{l_1 + 1}\,\delta_{k,l_1}\delta_{q,l_1+1}$ and

$$K_{kq,nm}(t) = \sum_{l_1,l_2=0}^\infty \Gamma_{11}(\omega_S, t)\sqrt{l_1 + 1}\sqrt{l_2 + 1}\,\delta_{k,l_1}\delta_{q,l_1+1}\delta_{m,l_2}\delta_{n,l_2+1}. \tag{B.5}$$

We perform the sum on $l_1, l_2$ and compute the associated Kossakowski matrix, that writes

$$\chi_{kq,nm}(t) = 2\operatorname{Re}\Gamma_{11}(\omega_S, t)\sqrt{k+1}\sqrt{m+1}\,\delta_{q,k+1}\delta_{n,m+1}. \tag{B.6}$$

Looking for a redefinition of the indices $(k, q) = i$ and $(n, m) = j$ as in Sec. 3.2, we notice that $n, q$ are forced to be equal to $m+1, k+1$ respectively. The only ordered couples with a nonzero contribution to the r.h.s. of (B.6) are of the form $(k, q) = (i, i+1)$, so that we can adopt the simple mapping $(k, q) = (i, i+1) \to j$ and $(m, n) = (j, j+1) \to i$. In this new notation Eq. (B.6) reads

$$\chi_{i,j}(t) = 2\operatorname{Re}\Gamma_{11}(\omega_S, t)\sqrt{i+1}\sqrt{j+1}. \tag{B.7}$$

This has an evident dyadic structure of the form $\chi_{i,j}(t) = 2\operatorname{Re}\Gamma_{11}(\omega_S, t)|a\rangle\langle a|$, where we indicated $|a\rangle = (1, \sqrt{2}, \sqrt{3}, \dots)$. The matrix above is positive semidefinite only if $\operatorname{Re}\Gamma_{11}(\omega_S, t) > 0$, which is guaranteed again by applying the second Markov approximation.

## C  Exact solution of the open three-level system

In this appendix we provide the exact solution of the open three-level system described in Sec. 4, which is originally described in Ref. [38]. Suppose that the initial state of the universe is the following pure state:

$$|\Psi(0)\rangle = (a_0(0)|0\rangle + a_1(0)|1\rangle + a_2(0)|2\rangle) \otimes |\Omega\rangle. \tag{C.1}$$

Since the total number of excitations is conserved, the state at time $t$ must be of the form

$$|\Psi(t)\rangle = (a_0(t)|0\rangle + a_1(t)|1\rangle + a_2(t)|2\rangle) \otimes |\Omega\rangle + \sum_p d_p(t)|0\rangle \otimes |1_p\rangle, \tag{C.2}$$

where $|1_p\rangle$ is the state of the bath that supports a single excitation at energy $\epsilon_p$. From here it is easy to see that the reduced density operator of the system can be written in the basis $\{|0\rangle, |1\rangle, |2\rangle\}$ as

$$\rho(t) := \mathrm{Tr}_{\mathcal{E}} |\Psi(t)\rangle\langle\Psi(t)| = \begin{bmatrix} 1 - |a_1(t)|^2 - |a_2(t)|^2 & a_0(t)a_1^*(t) & a_0(t)a_2^*(t) \\ a_0^*(t)a_1(t) & |a_1(t)|^2 & a_1(t)a_2^*(t) \\ a_0^*(t)a_2(t) & a_1^*(t)a_2(t) & |a_2(t)|^2 \end{bmatrix}. \tag{C.3}$$

Writing the evolution equation $i\partial_t |\Psi(t)\rangle = H_{\mathcal{U}} |\Psi(t)\rangle$ and comparing coefficients, one finds

$$\dot{a}_0(t) = 0, \tag{C.4a}$$

$$\dot{a}_\alpha(t) = -i\omega_\alpha a_\alpha(t) - i\sum_p g_{\alpha,p} d_p(t), \tag{C.4b}$$

$$\dot{d}_p(t) = -i\epsilon_p d_p(t) - ig_{1,p}a_1(t) - ig_{2,p}a_2(t), \tag{C.4c}$$

where $\alpha \in \{1, 2\}$. From Eq. (C.4a) we immediately conclude that $a_0(t) = a_0(0)$ for all $t \geq 0$. Remembering that $d_p(0) = 0$, Eq. (C.4c) can be formally integrated as

$$d_p(t) = -i\int_0^t d\tau\, e^{-i\epsilon_p(t-\tau)} \big[ g_{1,p}a_1(\tau) + g_{2,p}a_2(\tau) \big], \tag{C.5}$$

from which we obtain the following after substitution into Eq. (C.4b):

$$\dot{a}_1(t) = -i\omega_1 a_1(t) - \int_0^t d\tau\, c_{11}(t-\tau)a_1(\tau) - \int_0^t d\tau\, c_{12}(t-\tau)a_2(\tau), \tag{C.6a}$$

$$\dot{a}_2(t) = -i\omega_2 a_2(t) - \int_0^t d\tau\, c_{21}(t-\tau)a_1(\tau) - \int_0^t d\tau\, c_{22}(t-\tau)a_2(\tau), \tag{C.6b}$$

where $c_{\alpha\beta}$ is the correlation function in Eq. (33). This system can be solved with a Laplace transformation $\hat{f}(s) = \int_0^\infty dt\, f(t)e^{-st}$, after which

$$s\hat{a}_1(s) - a_1(0) = -i\omega_1 \hat{a}_1(s) - \hat{c}_{11}(s)\hat{a}_1(s) - \hat{c}_{12}(s)\hat{a}_2(s), \tag{C.7a}$$

$$s\hat{a}_2(s) - a_2(0) = -i\omega_2 \hat{a}_2(s) - \hat{c}_{21}(s)\hat{a}_1(s) - \hat{c}_{22}(s)\hat{a}_2(s). \tag{C.7b}$$

Now we impose the Lorentzian bath assumption (33)-(34): define $M := \mu + i\omega_0$ and the zero-determinant matrix $G_{\alpha\beta} := \gamma_{\alpha\beta}\mu/2$ and notice that $\hat{c}_{\alpha\beta}(s) = G_{\alpha\beta}/(p+M)$. Inserting above one obtains

$$\hat{a}_1(s) = \frac{a_1(0)[(s+i\omega_2)(p+M) + G_{22}] - G_{12}a_2(0)}{Q(s)}, \tag{C.8a}$$

$$\hat{a}_2(s) = \frac{a_2(0)[(s+i\omega_1)(p+M) + G_{11}] - G_{21}a_1(0)}{Q(s)}, \tag{C.8b}$$

where $Q(s) = s^3 + h_1 s^2 + h_2 s + h_3$ is a polynomial in $s$ with coefficients

$$h_1 = M + i(\omega_1 + \omega_2), \tag{C.9a}$$

$$h_2 = G_{11} + G_{22} - \omega_1\omega_2 + iM(\omega_1 + \omega_2), \tag{C.9b}$$

$$h_3 = -M\omega_1\omega_2 + i(\omega_1 G_{22} + \omega_2 G_{11}). \tag{C.9c}$$

Assuming that $Q(s)$ has three non-degenerate roots $r_1, r_2, r_3$ we can apply the following Lagrange partial fraction decomposition:

$$\frac{A(s)}{Q(s)} = \sum_{j=1}^3 \frac{A(r_j)}{Q'(r_j)}\frac{1}{s - r_j}, \tag{C.10}$$

where $Q'(s) = 3s^2 + 2h_1 s + h_2$ is the derivative of $Q(s)$. The result is

$$\hat{a}_1(s) = \sum_{j=1}^{3} \frac{a_1(0)[(r_j + i\omega_2)(r_j + M) + G_{22}] - G_{12}a_2(0)}{3r_j^2 + 2h_1 r_j + h_2} \cdot \frac{1}{s - r_j}, \tag{C.11a}$$

$$\hat{a}_2(s) = \sum_{j=1}^{3} \frac{a_2(0)[(r_j + i\omega_1)(r_j + M) + G_{11}] - G_{21}a_1(0)}{3r_j^2 + 2h_1 r_j + h_2} \cdot \frac{1}{s - r_j}. \tag{C.11b}$$

The inverse Laplace transform of these expressions leads to the desired solution:

$$a_1(t) = \sum_{j=1}^{3} \frac{a_1(0)[(r_j + i\omega_2)(r_j + M) + G_{22}] - G_{12}a_2(0)}{3r_j^2 + 2h_1 r_j + h_2} e^{r_j t}, \tag{C.12a}$$

$$a_2(t) = \sum_{j=1}^{3} \frac{a_2(0)[(r_j + i\omega_1)(r_j + M) + G_{11}] - G_{21}a_1(0)}{3r_j^2 + 2h_1 r_j + h_2} e^{r_j t}, \tag{C.12b}$$

together with $a_0(t) = a_0(0)$.

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
