# Peer review of "A time-dependent regularization of the Redfield equation"

_SciPost Physics, doi:SciPost Phys. 15, 117 (2023)_

## Round 1 · Referee Report · Anonymous · 2023-1-16

Report
The authors introduce a new regularisation of the weak-coupling Bloch-Redfield master equation for the dynamics of open quantum systems. Their regularisation preserves the complete positivity of the dynamical map generated by the master equation and it is shown to provide a very accurate description of the "true" dynamics of a 3-level open system.
The paper is written in a clear and pleasant way. Moreover, finding suitable and simple regularisations of the Bloch-Redfield equation is very important, as it is known that it's not always possible to apply the standard textbook secular approximation to the master equations for multipartite open quantum systems, especially in the presence of quasi-degeneracies. Therefore, the open system community is looking for master equations that are i) describing the open system dynamics in an accurate way; ii) mathematically well-defined (this is, e.g., crucial to study their quantum thermodynamics). The regularisation introduced in the manuscript is simple, elegant, and precise, so I believe it may be employed by the open system community in the future. It's also true that this regularisation is providing mostly a numerical method to fix the Bloch-Redfield equation, while no physical meaning of this procedure is given. Still, in my opinion the relevance of this method meets the criteria for acceptance in Scipost Physics, as having a completely positive and very accurate master equation will be very beneficial for numerical simulations of complex open quantum systems.
This being said, I have some important questions/comments before recommending the paper for publication:
1) There is something I can't understand about the example on the 3-level system. As we know, the Bloch-Redfield master equation (with or without the second Markov approximation) is obtained in the regime of weak coupling, otherwise applying the Born approximation is not possible. However, if I understood correctly, the example discussed by the authors is not in the weak coupling regime: in Figs. 2 and 3, $\gamma_1=\gamma_2=1$ and $\mu=20$, so the spectral density in Eq. (33) is not small compared to the system frequencies. Then, the exact solution of the dynamics (Appendix C) is still good, but the derivation of the Bloch-Redfield master equation (and thus of its regularisation) is ill-defined, as it does not correspond to the microscopic model anymore. Am I missing something here? If not, then the authors should present a new example in the correct regime of weak coupling, with the decay rates that are much weaker than the system frequencies. Indeed, it may be that the better accuracy of the regularised master equation with respect to the other ones shown in Figs. 2 and 3 is due to this ill-defined regime and it's of little use for practical purposes (e.g., this accuracy may change when the coupling is weak, even if mathematically the master equations are defined in the same way).
2) A comparison of different master equations with the exact open dynamics has been discussed in Hartmann and Strunz, Phys. Rev. A 101, 012103 (2020). There, a system of two qubits in a common bath is studied through an exact pseudo-mode mapping. Interestingly, it is shown that, for the sake of numerical precision, the mathematically ill-defined Bloch-Redfield equation is the most precise one, beating different approaches such as coarse-graining methods and partial secular approximation. In the present manuscript, the authors show that their regularisation is sometimes even more precise than the Bloch-Redfield approach. I am wondering how their regularisation behaves for the model studied by Hartmann and Strunz. I guess it should be possible to apply their method to this 2-qubit model without much effort. Then, I believe that beating the Bloch-Redfield accuracy also in this case would be very corroborating for their results and also extremely interesting for the community.
3) As I mentioned previously, the authors' regularisation is mostly a numerical recipe and gives no insights, for instance, on the quantum thermodynamics of this equation. May the authors make some comments about the steady state of this master equation? Is it still the Gibbs state of some modified Hamiltonian?
Some additional minor comments:
1) Right before Eq. (18), the authors distinguish between the Frobenius norm and the 2-norm. I thought these two norms were actually the same thing, so could the authors clarify this?
2) A typo: right before Eq. (11), "amounts in" instead of "amounts to".
3) The authors state in the introduction that their regularisation is remarkably valid also for time-dependent equations, contrary to many other examples in the literature. I would mitigate this claim, because most of the "regularised" master equations they refer to can be trivially extended to the time-dependent case as well.
Author: Antonio D'Abbruzzo on 2023-05-16 [id 3671]
(in reply to Report 1 on 2023-01-16)
We thank the Referee for their careful assessment of the manuscript and for thinking that our work is well-written and meets the criteria for acceptance in SciPost Physics. Below we address the remarks raised in the report while referencing the new version of the manuscript, which was amended following the Referee's suggestions.
Major comments 1. The weak-coupling approximation requires the coupling constants to be much smaller than the inverse of the environment correlation time. In the case of the three-level model studied here, this means that we assume $\gamma$ to be much smaller than $\mu$ (assuming for simplicity $\gamma_1 = \gamma_2 = \gamma$), and the system frequencies should not play any role in this separation of timescales. We acknowledge that this point should be clarified, hence we added appropriate comments below Eqs. (5) and (34). Moreover, we lowered the values of $\gamma$ in the plots by one or two orders of magnitude to reassure that we are working in the correct regime. 2. We greatly thank the Referee for pointing out this very interesting reference: we are certainly interested in investigating the performance of our approach on the Hartmann-Strunz model. However, given the amount of space it would take to discuss a new model here, we decided to postpone such analysis to a future work in which we are devising an even better strategy to numerically regularize the Redfield equation. Preliminary studies already suggest that positive results can be obtained also in this case. For now, we only added a citation to such work. 3. Studying the equilibrium steady state of the Redfield equation is known to present some subtleties. Specifically, one can show that the Gibbs state of the system Hamiltonian is not an exact steady state, in contrast to what happens with the secular-approximated master equation, and corrections should be included by considering a mean force Gibbs state, which is obtained by tracing the Gibbs state of the total Hamiltonian over the bath variables [see Lee and Yeo, Phys. Rev. E 106, 054145 (2022)]. Such calculations depend on the shape of the Kossakowski matrix, which we are modifying in an analytically unpredictable way (at least in the general case), thus making an immediate transition to the regularized scenario difficult. We thank the Referee for bringing up this point, but at this level we are not able to answer the question and further in-depth investigations are necessary.
Minor comments 1. The Frobenius norm is the norm associated with the Hilbert-Schmidt inner product, while the 2-norm is here intended as the Schatten 2-norm, which is the matrix norm induced by the vector Euclidean norm. The latter is also known as spectral norm and it can be obtained by taking the largest singular value. We acknowledge that the terminology is ambiguous: in the new version we added explicit definitions at the beginning of Sec. 3.2 and we substituted "2-norm" with "spectral norm". 2. We thank the Referee for spotting the typo, that has now been corrected. 3. Our claim of being capable of handling time dependence is related to the fact that we do not perform a "second Markov approximation", thus keeping the time dependence in the truncated one-sided Fourier transforms that appear in the definition of the Kossakowski matrix [see Eq. (10)]. In this scenario, the matrix is time dependent regardless of the presence of an external driving of the system Hamiltonian. While, as the Referee correctly pointed out, other regularized master equations can be extended to encompass the case in which the system Hamiltonian is time dependent, we are not aware of any immediate way to extend such equations to handle the time dependence in Eq. (10).
Author: Antonio D'Abbruzzo on 2023-05-16 [id 3672]
(in reply to Report 2 on 2023-04-14)We thank the Referee for their insightful comments and for the general positive evaluation, recommending publication in SciPost Physics. Below we address the remarks raised in the report, also describing how they affected the new version of the manuscript.
1. We thank the Referee for this remark. Colors and styles of the plots have been changed to improve readability.
2. We greatly thank the Referee for the suggestion, since it allowed us to discover an additional point about our approach which is now appropriately described at the end of Sec. 4.2. We found that the performance of our method changes according to the ratio between spectral width $\mu$ and frequency range of the system $\omega_R = \max (\omega_1, \omega_2)$. When $\mu \gg \omega_R$ the Kossakowski matrix is essentially positive and our procedure has little effect on the dynamics [see the discussion below Eq. (25)]. By lowering $\mu$ we start to see the benefits of our approach, surpassing in performance the other master equations. However, when $\mu < \omega_R$ the Kossakowski matrix starts to be sensibly non-positive and our procedure is less accurate than Redfield itself. Our interpretation here is that the non-Markovianity features cannot be ignored in this regime and the truncation of the negative part has a too drastic effect. We already envisioned a way to improve our approach and avoid this shortcoming by including non-Markovian effects in the regularization: it will be presented in a future work. For now, the regularization presented here works best when the Kossakowski matrix is only slightly non-positive, in accordance with our statement of searching for a CP-divisible dynamics.
3. By construction, our approach is expected to be valid whenever it is justified to use a (Markovian limit of the) Redfield equation, and the accuracy of the latter does not directly depend on the interaction strength among constituents of the system. The local vs global debate becomes relevant when one enforces an additional secular approximation, which can jeopardize the general validity of a global approach. See for example Cattaneo et al., New J. Phys. 21, 113045 (2019), where the authors show that global equations are expected to be generally more appropriate than their local counterparts, provided non-secular terms are treated accordingly and not discarded. Since our regularization is not secular-like, here we are adopting a similar view: by choosing a local basis instead of a global one we would introduce unnecessary additional errors.
4. The Referee is of course correct in thinking that the negative eigenvalue of the Kossakowski matrix could be used as a measure of non-Markovianity. In fact, as shown in Hall, Cresser, and Andersson, Phys. Rev. A 89, 042120 (2014), this approach is essentially equivalent to the well-known Rivas-Huelga-Plenio measure introduced in Rivas, Huelga, and Plenio, Phys. Rev. Lett. 105, 050403 (2010). In the new version of the manuscript we added a comment to highlight this connection below Eq. (25), with appropriate references.
5. We thank the Referee for spotting the typo, that has now been corrected.

---

## Round 1 · Referee Report · Anonymous · 2023-4-14

Report
In this work, the authors propose a novel technique to compute/ derive a positive Kossakowski matrix, ensuring a valid GKSL equation for the time-evolution of an open quantum system. Whereas traditional assumptions (Markov and secular approximations) leading to that type of equations are easy to satisfy and verify for few paradigmatic models of open quantum systems (single-, two- and three-level quantum systems, non-degenerate, with suitable spectral density of the baths compared to the energy scales of the system), it is in general a complex task to derive a valid GKSL equation for an open quantum system as the Kossakowski matrix can in general be non-positive.
Starting from the Bloch-Redfield equation and the corresponding Kossakowski matrix χ(t), they put forward a procedure to find a positive matrix that is the closest one from χ(t). Interestingly, they claim that their procedure remains valid in the transient regime, i.e. accounting for an additional time-dependence.
This work addresses a timely problem in the broad field of open quantum systems, and proposes an interesting and novel way to find a correct solution for finding a valid GKSL equation. The authors illustrate their procedure with a simple but non-trivial example, a three-level model and demonstrate the validity of their scheme compared to the exact solution and other valid approaches. The paper is very well-written, in a clear and rigorous way, with analytical calculations and numerical results. I believe it definitely deserves publication in SciPost. Prior to it, I would have few questions and comments.
1) Figure 2 does not read very well when printed in B&W. One option to improve it could be to highlight the exact solution and the one obtained from the regularized Redfield procedure in dotted and dashed lines. Same for Fig. 3.
2) The authors illustrate their procedure with a three-level model and a Lorentzian spectral density function for the bath assumed to be flat over the typical frequencies of the system (see discussion below Eq. 42). This is of course justified to ensure a Markovian evolution, and provides a valid benchmark with the exact solution. However, the Kossakowski matrix may become non-positive when the Lorentzian spectral density is not flat enough for instance. As the novel regularization scheme is a general approach, it would be interesting and important to also treat this situation. Could the authors compare their solution with the exact solution, showing that their regularization scheme works, while other techniques may break down?
3) A situation that has triggered several questions and works in the past years concerns the derivation of a valid GKSL equation in the case of two or more interacting quantum systems, weakly coupled to environments. Depending on the interaction strength between the constituents, it was shown that a quantum master equation in the local computational basis or in the energy eigenbasis was more accurate. In this work, the three-level model is described in its energy-eigenbasis, avoiding this question. Could the authors comment how their regularization procedure may depend on the interaction strength among the constituents of the quantum system? Do the authors expect any advantages of their approach to interpolate between different interaction regimes?
4) Interestingly, the authors explain that the variance in Eq. (25) that depends on the width of the correlation functions of the bath will determine how far or close the regularized matrix will be from the initial one. I was wondering whether lambda_- for instance, the contribution that is disregarded, could be used as a measure for non-Markovianity (which is in general not trivial to define)? Could the authors comment on that?
5) Possible typo: sentence below Eq. (18), a word is possibly missing “[…] it is sufficient a spectral decomposition”.

---

## Round 2 · Referee Report · Anonymous · 2023-6-12

Report
The authors have addressed my requests in a convincing way. They have not performed further numerical studies following Hartmann and Strunz as I was suggesting, but I understand there is already a lot of material in the current version of the paper. Moreover, I really like the new figure 3, which partially investigates the trade-off between Redfield and regularised master equation.
I would like to recommend the paper for publication after the following two issues are discussed:
i) I kind of disagree with the authors' assertion about the weak-coupling limit being based on $\tau_\epsilon\ll \tau_S$ only (memory time of the environment being much smaller than the system relaxation time). This is the necessary condition for performing the "first Markov" approximation. However, all the derivations of the master equations I am aware of make use of a perturbative treatment that requires that the system-environment interaction is a perturbation of the total Hamiltonian. That is, the system-bath coupling is much smaller than the system energy ($\gamma \ll \omega_{1,2}$). This is discussed, for instance, in the standard textbook by Breuer & Petruccione. A different derivation based on Nakajima-Zwanzig expansion can also be found in the textbook by Rivas and Helga on open quantum systems. Note that this condition does not immediately imply $\tau_\epsilon \ll \tau_S$, and vice versa. For instance, very high temperatures and cutoff frequencies in the Caldeira-Leggett model (see Section 3.6 of Breuer and Petruccione) can lead to very small $\tau_\epsilon$ even if the coupling constant is large. So, if the authors are not basing their weak-coupling analysis on some different arguments that I currently missing, I believe they should modify the justification of the weak coupling limit in the text. I guess they do not need to modify the values in the numerical experiments, because the weak coupling limit is already kind of satisfied. To be fair, 0.3 in figure 2 is not much smaller than 1, but it is still acceptable in the present context (figure 3 is totally fine).
ii) The authors have correctly replied that right now they cannot say anything about the steady state of the regularised equation and they have made some interesting connections with the mean-force Gibbs state. I would really appreciate if they added these comments to the manuscript, because I believe they can be of interest for a broader audience.
Author: Antonio D'Abbruzzo on 2023-07-13 [id 3807]
(in reply to Report 1 on 2023-06-12)We thank the referee for the additional comments. Below we describe how the two remarks affected the new version of the manuscript.
We modified our description of the weak-coupling limit by adopting the following milder formulation: the inverse coupling constant provides the longest timescale of the problem. We hope this is sufficient to rule out further concerns about the applicability region of the master equations discussed here. A citation to Mozgunov and Lidar, Quantum 4, 227 (2020) was added below Eq. (5), where the question is addressed more thoroughly.
We thank the referee for finding the connection with the mean force Gibbs state interesting. As requested, we added a couple of sentences in the Conclusions section to briefly describe what was said during our correspondence. A citation to Lee and Yao, Phys. Rev. E 106, 054145 (2022) was therefore added.

---

## Round 2 · List of Changes

- Comments about the weak-coupling regime were added below Eqs. (5) and (34).
- In Sec. 3.2, "2-norm" was replaced by "spectral norm", and definitions of the involved norms were added.
- A citation to Hartmann and Strunz, Phys. Rev. A 101, 012103 (2020) was added (now Ref. [23]).
- Colors and styles of the plots were changed to improve readability.
- In numerical calculations, the value of the coupling constant was lowered.
- A discussion about the role of the spectral width was added in Sec. 4.2, and Fig. 3 was updated appropriately.
- A link to the RHS non-Markovianity measure was added below Eq. (25), with appropriate references.
- Typos were corrected.

---

## Round 3 · List of Changes

• The statement of weak-coupling regime was modified below Eq. (5), and a citation to Mozgunov and Lidar [20] was added.
  • In the Conclusions section, the open problem of the thermodynamic characterization of the regularized Redfield equation was specified, together with a brief discussion on the mean force Gibbs state for the traditional Redfield equation. Correspondingly, a citation to Lee and Yeo, Phys. Rev. E 106, 054145 (2022) was added.
  • Typos were corrected.

---

## Editorial Decision

published